# Unleashing the Power of Language Models in Text-Attributed Graph

**Haoyu Kuang[1], Jiarong Xu[2]\*, Haozhe Zhang[5], Zuyu Zhao[5],**
**Qi Zhang[3], Xuanjing Huang[3], Zhongyu Wei[1, 4]\***

[1]School of Data Science, Fudan University, China
[2]School of Management, Fudan University, China
[3]School of Computer Science, Fudan University, China
[4]Research Institute of Intelligent Complex Systems, Fudan University, China
[5]Huawei Technologies Co.,Ltd, China

hykuang23@m.fudan.edu.cn, {jiarongxu, qz, xjhuang, zywei}@fudan.edu.cn
{zhanghaozhe7, zhaozuyu1}@huawei.com

## Abstract

Representation learning on graph has been demonstrated to be a powerful tool for solving real-world problems. Text-attributed graph carries both semantic and structural information among different types of graphs. Existing works have paved the way for knowledge extraction of this type of data by leveraging language models or graph neural networks or combination of them. However, these works suffer from issues like underutilization of relationships between nodes or words or unaffordable memory cost. In this paper, we propose a **N**ode **R**epresentation **U**pdate **P**re-training Architecture based on Co-modeling Text and Graph (**NRUP**). In NRUP, we construct a hierarchical text-attributed graph that incorporates both initial nodes and word nodes. Meanwhile, we apply four self-supervised tasks for different level of constructed graph. We further design the pre-training framework to update the features of nodes during training epochs. We conduct the experiment on the benchmark dataset ogbn-arxiv. Our method achieves outperformance compared to baselines, fully demonstrating its validity and generalization.

## 1 Introduction

Text-attributed graphs, characterized by the association of nodes with text attributes (Yang et al., 2021), are prevalent in diverse real-world contexts. For instance, in paper citation networks, each paper is accompanied by textual content, while in social networks, each user can be described through a text description. The investigation of learning techniques on text-attributed graphs has garnered considerable attention in domains such as graph learning, information retrieval, and natural language processing, reflecting the growing importance of understanding and analyzing textual information within the context of graph-based structures.

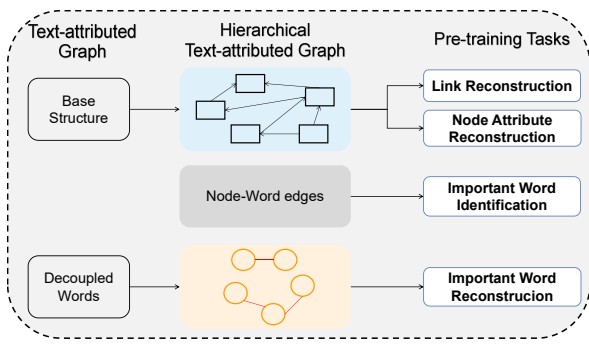

Figure 1: An illustration of hierarchical text-attributed graph and corresponding self-supervised tasks for different level.

Existing research on learning from text-attributed graph mainly falls into three lines: (1)Language models(LMs) only, in these works, LMs(Kim, 2014; Vaswani et al., 2017) are applied to leverage the local textual information of individual nodes and generate representation for them(Howard and Ruder, 2018). However, structural relationship between nodes are ignored in this way. To leverage the relationships between nodes, an self-supervised learning framework GIANT(Chien et al., 2021) propose to guide the training of LM with graph structure. Nevertheless, LM-based methods ignore the message passing among nodes; (2)Graph neural networks(GNNs) only(Kipf and Welling, 2017; Zhang and Chen, 2018), which are used to capture the structural properties of text-attributed graphs. Raw texts contained by each node are transformed to numerical features as node attributes(Hu et al., 2020a; Liu et al., 2020; Hu et al., 2020c) by graph-irrelevant methods, such as bag-of-words, pre-trained bert in most previous studies. Clearly, the information contained in raw text is compressed and underutilized. Additionally, relying solely on a fixed

---
\*Corresponding author

representation of text may not be appropriate for certain scenarios. For instance, the term 'Transformer' can refer to a device used for adjusting the voltage of an electric power supply, while in an academic context, it signifies a specific model; (3)combination of LMs and GNNs(Bi et al., 2021; Zhu et al., 2021a), which boost the text embedding with graph structure. However, it suffers from severe scalability issues when facing with large-scale graph and huge parameters of LMs. To address this, GLEM (Zhao et al., 2023) leverages a variational EM framework to iteratively update both the LM and GNN modules, enabling scalability to real-world graphs. Nevertheless, GLEM relies on task-specific labels, resulting in node representations that are constrained to the specific task at hand. Generally, prior researches encounter issues such as overlooking the relationships between nodes or words, scalability limitations, and a lack of generalizability.

In this paper, we introduce a general text-attributed graph pre-training framework that could fully utilize the relationship between graph-based structure and textual information. The main contributions of our proposed research are as follows.

First, to enhance the modeling of textual information within nodes of text-attributed graphs, we construct a hierarchical text-attributed graph that incorporates both initial nodes and word nodes. More specifically, we further decouple the word nodes from the corpus consisting of the textual information from all nodes. Then we construct edges among nodes based on word occurrence in nodes (node-word edges) and word co-occurrence in the whole corpus (word-word edges), as shown in figure 1. This enables us to capture the finer nuances of the text at a more granular level.

Second, approaching the capability of generating effective representations adapted to various scenarios, we introduce a multi-task graph pre-training framework. This framework encompasses various self-supervised tasks, such as link reconstruction, node attribute reconstruction, important word reconstruction, and important word identification. The objective of link reconstruction is to capture the underlying structural patterns in a general sense, while node attribute reconstruction aims to uncover the semantic relationships among nodes. Furthermore, the tasks of important word reconstruction and important word identification are specifically designed for access to distinctive semantics and

paper-occurrence correlation, respectively.

Third, to mutually boost representations of nodes and words, we employ a relational graph neural network (R-GNN) as the foundational model for acquiring knowledge from the hierarchical text-attributed graph. Furthermore, within our framework, we introduce two aggregators that iteratively refine the features of both nodes and words, leveraging progressively optimized embeddings of papers/words after a designated number of training epochs.

## 2 Method

In this section, we present the entire training framework to learn paper and word representations simultaneously without supervision based on R-GNNs, including the modeling method of text-attributed graph, self-supervised tasks and pre-training architecture, see overall framework in figure 2.

### 2.1 Hierarchical Text-attributed Graph

To better establish the relationship between the raw text and the graph, we propose to construct a hierarchical text-attributed graph encompassing initial nodes and word nodes, inspired by Yao et al. (2019).

First, we tokenize the raw text contained in the nodes, thereby acquiring all the individual words;

Second, we construct a hierarchical text-attributed graph that incorporates both initial nodes and word nodes.

Third, we construct edges among nodes for the hierarchical text-attributed graph. The relationship between initial nodes constitutes the edges between them; and we build paper-word edges based on word occurrence in papers; as for edges between word nodes, we employ point-wise mutual information (PMI) to measure the co-occurrence frequency between words to determine whether to build a edge. The PMI value is computed as follows:

$$PMI(i,j) = \log \frac{p(i,j)}{p(i)p(j)}$$
$$p(i,j) = \frac{W(i,j)}{W}$$
$$p(i) = \frac{W(i)}{W}$$

where $i, j$ represents two word nodes; $W(i)$ is the number of sliding windows in the nodes that contain word $i$; $W(i,j)$ is the number of sliding windows in the nodes that contain both word $i$ and $j$; and $W$ is the total number of sliding windows in the corpus from all nodes. We construct edges

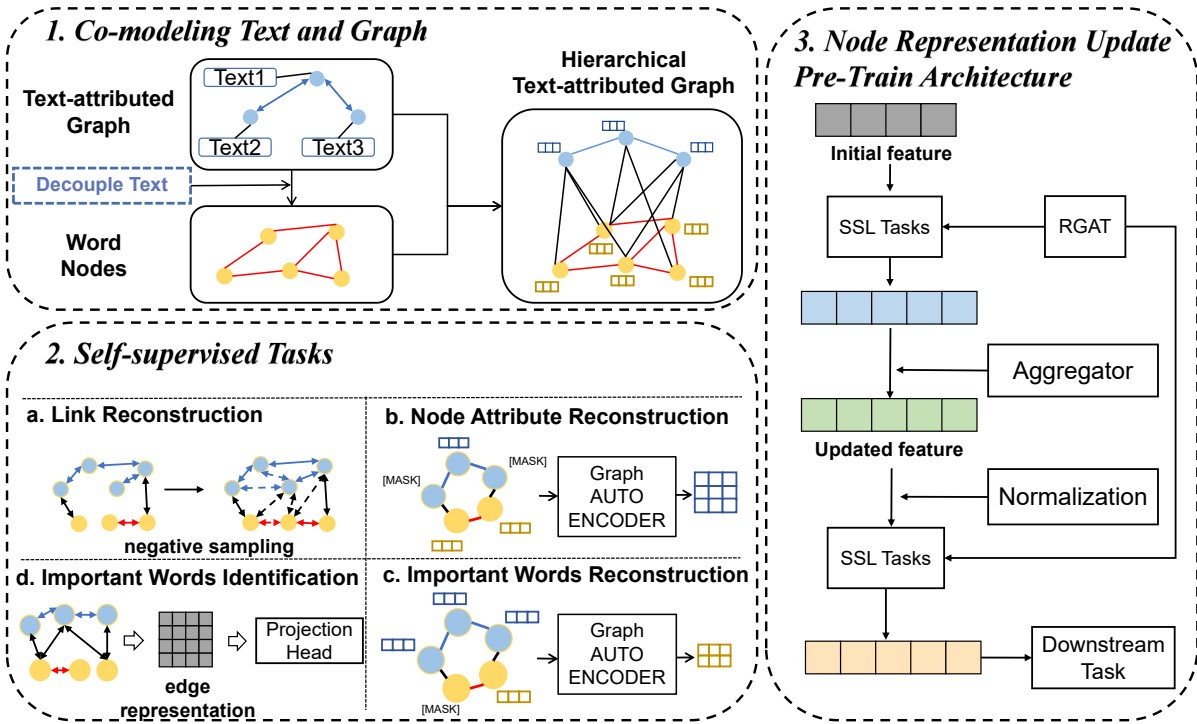

Figure 2: An illustration of of the entire process of **NRUP**. 1). Co-modeling the text and graph based on heterogeneous graph. 2). Self-supervised tasks as training objectives. 3). Pre-training architecture updates the features with aggregators.

for words with positive PMI value which suggests a high semantic correlation of words.

Thus far, we have constructed the structure of hierarchical text-attributed graph. Next, we need to assign corresponding node attributes to different types of nodes. Initially, we use a pre-trained Bert model to generate word embedding for each word node as the attributes; Subsequently, we simply average the embeddings of all word nodes in one node to obtain the feature of initial nodes. Finally, we obtain this hierarchical text-attributed graph with semantic and structural information coexisting.

Let $\mathcal{G} = (\mathcal{V}, \mathcal{E}, \mathcal{J}, \mathcal{K}, \varphi, \phi, \mathcal{X}, \mathcal{Z})$ denotes the hierarchical text-attributed graph we built, where $\mathcal{V}$ and $\mathcal{E}$ represent the sets of nodes and edges respectively; $\mathcal{J}$ and $\mathcal{K}$ represent sets of node types and edge types respectively; $\varphi : \mathcal{V} \rightarrow \mathcal{J}$ is the node type mapping function, while $\phi : \mathcal{E} \rightarrow \mathcal{K}$ is the edge type mapping function; $\mathcal{X}^{n \times d}$ and $\mathcal{Z}^{m \times d}$ represent the feature matrix of paper and word nodes respectively; $n$ and $m$ denote the number of initial nodes and words respectively, and $d$ denotes the feature dimension. The number of nodes $|\mathcal{V}|$ is the summation of the number of initial nodes and individual words.

Intuitively, by constructing hierarchical text-attributed graph with initial nodes and words, we build a bridge for information interaction. Consequently, as we incorporate individual words into the training process, the initial nodes are endowed with knowledge from both interconnected nodes and their respective textual components.

## 2.2 Self-supervised Tasks

Appropriate tasks drive R-GNNs to mine potential structure and semantic information in heterogeneous graph continuously. In our pre-training architecture, we apply 4 tasks to fully mine the information of different level in hierarchical text-attributed graph, as shown in figure 1.

### 2.2.1 Link Reconstruction

Briefly speaking, link reconstruction is to predict existing edges between node pairs.

**Task Process**: In the design of link reconstruction task, we regard it as a binary classification problem, and train the model by negative sampling.

First, we treat the edges existing in the graph as positive samples, and sample some non-existent edges in the graph as negative samples.

Second, for each node pair $(u, v)$ in the graph, we calculate their score: $e_{u,v} = \varphi(h_u, h_v)$ based on the representation $h_u$ and $h_v$, where $\varphi$ is a dot

product, and can be any other function that computes similarity.

Third, labeling the positive sample as 1 and the negative sample as 0, we can optimize R-GNN with the following loss function:

$$\mathcal{L}_{LR} = -\log \sigma(e_{u,v}) - \sum_{v_i \sim P_n(v)} \log(1 - \sigma(e_{u,v_i}))$$

where $\sigma$ denotes a activation function, and $v_i \sim P_n(v)$ denotes the negative sampling distribution.

We perform link reconstruction for edges among initial nodes.

### 2.2.2 Node Attribute Reconstruction

In our work, we use node attribute reconstruction task to maximize the semantic information of hierarchical text-attributed graph, also known as feature reconstruction in homogeneous graphs(Hou et al., 2022).

**Task Process**: First, we sample a subset $\tilde{\mathcal{V}}_{initial} \subset \mathcal{V}_{initial}$, and mask their features with a mask token [MASK], i.e., a learnable vector $x_{[M]} \in {}^d$, and the feature matrix of word nodes remains unchanged. Thus the node feature $\tilde{x}_i$ can be defined as:

$$\tilde{x}_i = \begin{cases} x_{[M]}, v_i \in \tilde{\mathcal{V}}_{paper} \\ x_i, v_i \notin \tilde{\mathcal{V}}_{paper} \end{cases}$$
$$\tilde{z}_i = z_i, v_i \in \mathcal{V}_{word}$$

Second, we input the feature matrix $\tilde{\mathcal{X}}, \tilde{\mathcal{Z}}$ and graph $\mathcal{G}$ into a graph encoder $f_e$ to obtain the latent code $H$. Then we replace $H$ on masked node indices again with another mask token [DMASK].

Third, input the re-masked code matrix $\tilde{H}$ into a decoder $f_d$ to obtain the reconstructed feature matrix $W$. Then optimize with the scaled cosine error:

$$\mathcal{L}_{NAR} = \frac{1}{\left|\tilde{\mathcal{V}}_{paper}\right|} \sum_{v_i \in \tilde{\mathcal{V}}_{paper}} (1 - \frac{x_i^T w_i}{\|x_i\| \cdot \|w_i\|})^\gamma$$

which is averaged loss over all masked initial nodes. The scaling factor $\gamma$ is a hyper-parameter.

### 2.2.3 Important Word Reconstruction

Intuitively, we believe that there are important words in the raw text of initial nodes which can reflect the semantic information of the node to a large extent. Motivated by this, we design important word reconstruction task to reconstruct semantic information of important words.

**Task Process**: We need to define what are important words to the initial nodes. For instance, the

title of a paper contains key information thus we remove the stop words after tokenizing the title of the paper, and take the remaining words as important words. We mask the features of these important words and reconstruct them. The reconstruction loss is denoted as:

$$\mathcal{L}_{IWR} = \frac{1}{\left|\tilde{\mathcal{V}}_{IM}\right|} \sum_{v_i \in \tilde{\mathcal{V}}_{IM}} (1 - \frac{z_i^T w_i}{\|z_i\| \cdot \|w_i\|})^\gamma$$

### 2.2.4 Important Word Identification

Each node contain distinctive important words. Therefore, we design important word identification task with the objective of judging important words.

**Task Process**: First, we label the paper-word edge differently according to whether it is a important word to this node, 1 for important and 0 for unimportant.

Second, we splice the representation $h_u$ and $h_m$ of each node-word pair $(u, m)$ as the edge representation $h_{u,m}$.

Third, we input the edge representation into a projection head to predict the edge label, then optimize with the following loss function:

$$h_{u,m} = h_u \oplus h_m$$
$$y'_{u,m} = projection - head(h_{u,m})$$
$$\mathcal{L}_{IWI} = \sum_{u,m} -\log \sigma(y'_{u,m}) - \log(1 - y'_{u,m})$$

where $y'_{u,m}$ denotes the probability of being predicted as an important word edge.

### 2.3 Pre-training Framework

The representation of initial nodes and word nodes can be optimized simultaneously based on hierarchical text-attributed graph. Motivated by this, we propose a pre-traininig framework: Node Representation Update Pre-training Architecture(NRUP).

In our framework, we use the constructed graph as the input; and we select a R-GNN model as our base model, then the combination of self-supervised tasks is served as the objectives of the pre-training stage. Furthermore, we design two *aggregators* to update features of both initial nodes and words, which are denoted as follows:

$$Aggregator_{initial} \leftarrow MEAN - AGG(v, neigh_v^{initial}) + c * MEAN - AGG(neigh_v^{word})$$

$$Aggregator_{word} \leftarrow MEAN - AGG(n, neigh_n^{word}) + d * MEAN - AGG(neigh_n^{initial})$$

$$c = \left| neigh_{initial}^{initial} \right| / (\left| neigh_{initial}^{initial} \right| + \left| neigh_{initial}^{word} \right|)$$
$$d = \left| neigh_{word}^{word} \right| / (\left| neigh_{word}^{initial} \right| + \left| neigh_{word}^{word} \right|)$$

where $MEAN - AGG$ denotes a mean-aggregator, which average the embeddings of the aggregated nodes; $neigh_v^{initial}$ denotes 1-hop paper-neighbors of initial node $v$; while $neigh_n^{word}$ denotes 1-hop word-neighbors of word node $n$; $c$ and $d$ are **adaptive parameters** based on the average neighbor count of papers and words respectively, $\left| neigh_{initial}^{word} \right|$ denotes the number of word neighbors of the initial node, and $\left| neigh_{word}^{initial} \right|$ denotes the number of initial node neighbors of the word node.

After aggregation, we normalize the aggregated initial node and word embeddings separately to obtain the updated embedding matrix $U_{initial}^{n \times d}$ and $U_{word}^{m \times d}$. Then we replace the initial node features with the updated matrix, and continue to train the same R-GNN with self-supervised tasks until convergence.

## 2.4 Multi-Task Pre-training

Link reconstruction task tends to restore structural features, while node attribute reconstruction task focuses on semantic information. IWI and IWR tasks focus on deep mining of important words. Therefore, we combine these tasks to guide the representation learning of the R-GNN model. Loss function $\mathcal{L}$ can be denoted as:

$$\mathcal{L} = \mathcal{L}_{NAR} + \lambda_1 \mathcal{L}_{LR} + \lambda_2 \mathcal{L}_{IWI} + \lambda_3 \mathcal{L}_{IWR}$$

where $\lambda_1, \lambda_2, \lambda_3$ are hyper-parameters.

## 3 Experiment Setup

In this section, we apply our entire pre-training method to a real-world citation network *ogbn-arxiv*(Hu et al., 2020a) and report performance on the downstream tasks. To demonstrate the generalization of our method, we considered two settings on the same dataset: **Transductive Learning** and **Inductive Learning**. Meanwhile, we select several baseline models to prove the validity of our method.

### 3.1 Dataset

**Data Profiling**: The ogbn-arxiv dataset is a benchmark node classification dataset, representing the citation network between all Computer Science (CS) arXiv papers indexed by MAG. Each node with its raw text of title and abstract is an arXiv paper and each directed edge indicates that one paper cites another one. In addition, all papers are also associated with the year that the corresponding paper was published.

**Downstream Tasks**: We evaluate our model on two types of tasks, namely subject prediction and important words identification.

- *Subject Prediction*: This task is regarding prediction of subject areas of arXiv CS papers, which are manually labeled by the paper's authors and arXiv moderators. Formally, the task can be formulated as a 40-class classification problem.
- *Important Words Identification*: This task is designed to identify important words based on paper-word correspondence. Formally, it can be regarded as a binary classification problem.

### 3.2 Pre-Training and Fine-Tuning Setup

Basically, pre-training methods are designed to obtain the transferable knowledge from unlabeled datasets, thus pre-training models bring better representations for the downstream tasks. Therefore, to evaluate the effectiveness of our method, we propose two different setups.

The first setting is called **Transductive Learning**, we pre-train and fine-tune on the same graph in this setting, which means all nodes are visible during both pre-traing and fine-tuning stage. The second one is called **Inductive Learning**, we pre-train on one grpah and fine-tune on another graph in this setting. Generating representation for unseen nodes in fine-tuning stage makes this setting more challengeable.

| Dataset | Nodes | Edges | Avg.degree | Split Ratio |
|---|---|---|---|---|
| Ogbn-arxiv | 169343 | 1166243 | 13.7 | 41/13/17/29 |

Table 1: Data Split of Ogbn-arxiv

We propose to split the dataset into four parts as shown in Table 1 based on the publication dates of the papers to adapt to the pre-training settings, where 41% of papers published before 2017; 13% of papers published in 2017; 17% of papers published in 2018; 29% of papers published since 2019. Specifically, the descriptions of the two settings are as follows:

- *Transductive Learning*: Under this setting, we select all papers to involve in the pre-training stage; In the fine-tuning stage, we propose to

| Setting | Graph | Papers | Words | Paper-Paper edge | Paper-Word edge | Word-Word edge |
|---|---|---|---|---|---|---|
| Transductive Learning | Pre-Train/Fine-Tune | 169343 | 17634 | 2484941 | 21175983 | 7286638 |
| Inductive Learning | Pre-Train | 69499 | 17634 | 534337 | 8359451 | 3634502 |
| | Fine-Tune | 99844 | 17634 | 927020 | 12816390 | 4081080 |

Table 2: Constructed Hierarchical Text-attributed Graph

| Method | Subject Prediction Accuracy(%) | | Important Words Identification ROC-AUC | |
|---|---|---|---|---|
| | Transductive | Inductive | Transductive | Inductive |
| Feat | $59.17 \pm 0.06$ | $60.05 \pm 0.01$ | $0.7463 \pm 0.0012$ | $0.7715 \pm 0.0008$ |
| Attribute Masking | $65.32 \pm 1.98$ | $61.96 \pm 2.48$ | $0.7405 \pm 0.0055$ | $0.7752 \pm 0.0052$ |
| Edge Prediction | $64.94 \pm 2.04$ | $62.46 \pm 2.08$ | $0.7394 \pm 0.0072$ | $0.7736 \pm 0.0049$ |
| DGI | $70.34 \pm 0.16$ | $63.66 \pm 0.04$ | $0.7473 \pm 0.0062$ | $0.7707 \pm 0.0014$ |
| GPT-GNN | $68.45 \pm 2.54$ | $66.04 \pm 2.09$ | $0.7403 \pm 0.0078$ | $0.7689 \pm 0.0111$ |
| GraphMAE | $71.75 \pm 0.17$ | $67.42 \pm 0.35$ | $0.7475 \pm 0.0032$ | $0.7812 \pm 0.0009$ |
| GIANT | $\mathbf{72.46 \pm 0.07}$ | $68.89 \pm 0.06$ | $0.7433 \pm 0.0008$ | $0.7725 \pm 0.0012$ |
| NRUP | $72.33 \pm 0.14$ | $\mathbf{69.67 \pm 0.12}$ | $\mathbf{0.7515 \pm 0.0011}$ | $\mathbf{0.7847 \pm 0.0023}$ |

Table 3: Main result of Subject Prediction and Important Words Identification; we report Accuracy in task subject prediction and ROC-AUC in task important word indentification(bolded number is the best in that column).

train on papers published until 2017, validate on those published in 2018, and test on those published since 2019.

- *Inductive Learning*: Under this setting, we propose to pre-train on papers published until 2016, train on those published in 2017, validate on those published in 2018, and test on those published since 2019.

For the evaluation protocol, we conduct the same experimental process under two settings. First, we train a R-GNN encoder by the proposed **NRUP** based on pre-train graph. Then we freeze the parameters of the encoder and generate all the nodes' embeddings for fine-tune graph. For evaluation, we train a linear classifier and report the mean and standard deviation of performance on the test nodes through 10 random initializations.

### 3.3 Implementation Details

**Construction of Hierarchical Text-attributed Graph**: We construct pre-train and fine-tune hierarchical text-attributed graphs based on ogbn-arxiv according to two settings as shown in Table 2. The process time for construction can be found in the Appendix E.

**Basic Settings**: In NRUP, R-GAT is selected as the base model. We update the features of both papers and words with aggregator after training for 2000 epochs, then we input normalized updated features into the same R-GAT. Throughout the entire pro-

cess, we train the model to minimize the loss $\mathcal{L}$ using AdamW Optimizer and cosine learning rate decay without warmup. We provide an explanation of the hyper-parameter settings for different losses in Appendix D. More details and hyper-parameters can be found in Appendix A.

### 3.4 Baseline Models

To verify the effectiveness of our method, we select several baseline models for comparison.

- **Feat**: Fixed representation of paper generated by the BERT model.
- **Attribute Masking**: Mask the attributes of some nodes in the graph and reconstruct masked attributes by projection head.
- **Edge Generation**: Mask some edges and generate them based on nodes and remaining edges, and optimize the by contrastive loss.
- **DGI**(Velickovic et al., 2019): Maximizing mutual information between patch representations and corresponding high-level summaries of graphs.
- **GPT-GNN**(Hu et al., 2020c): A generative pre-training framework, which trains the model by reconstructing node attributes and graph structure through joint optimization of attribute generation and edge generation.
- **GraphMAE**(Hou et al., 2022): A generative self-supervised learning framework, which reconstructs initial node features by masked

graph autoencoder, and the model is optimized by reconstruction loss.

- **GIANT**(Chien et al., 2021): An SSL framework that generates numerical node features with graph-structured self-supervision by XR-Transformer.

# 4 Experiment Results

## 4.1 Main Results

Tabel 3 presents the performance of applying different pre-trian methods on the same pre-train dataset and fine-tune test set.

In task **Subject Prediction** we predict subject of paper based on paper representation, and in task **Important Words Identification** we identify important words based on concatenation representation of paper and word. More details can be found in Appendix B.

In both *Transductive Learning* and *Inductive Learning* setting, our NRUP achieves better or competitive performance compared to the selected baseline models, demonstrating the effectiveness and transferability of our method.

## 4.2 Effect of Different Tasks

The experimental results demonstrate that using multi-task loss for optimization can help the model capture both semantic and structural information in heterogeneous graph simultaneously.

We further investigated the performance on the test dataset using different tasks under *Inductive Learning* setting without embedding update. Tabel 4 shows the results that scenario-specific tasks can bring benefits to basic tasks, and our NRUP with multi-loss achieve the best performance.

| Self-Supervised Task | Accuracy |
|---|---|
| Node Attribute Reconstruction(NAR) | $66.65 \pm 0.14$ |
| Link Reconstruction(LR) | $65.98 \pm 0.11$ |
| NAR+LR | $67.31 \pm 0.07$ |
| NAR+IWI | $66.77 \pm 0.12$ |
| NAR+IWR | $66.85 \pm 0.04$ |
| LR+IWI | $66.10 \pm 0.11$ |
| LR+IWR | $66.29 \pm 0.09$ |
| NRUP | $\mathbf{67.56 \pm 0.03}$ |

Table 4: Effect of Different Tasks

## 4.3 Optimized Word Embedding

The optimization of word's embedding is a characteristic of our co-modeling method based on hier-

archical text-attributed graph. We conduct experiment under *Inductive Learning* setting using two basic self-supervised tasks to verify that embeddings of words have indeed been optimized.

| Self-Supervised Task | Accuracy |
|---|---|
| Bert Embedding | $60.05 \pm 0.01$ |
| Word Embedding of NAR | $66.18 \pm 0.01$ |
| Paper Embedding of NAR | $67.32 \pm 0.04$ |
| Word Embedding of LR | $67.92 \pm 0.04$ |
| Paper Embedding of LR | $66.87 \pm 0.10$ |

Table 5: Optimized Word Embedding (Word embedding means the average representation of word nodes, Paper embedding means the node representation output by RGAT)

We average the optimized word representation obtained by training for 2000 epochs through a certain task as the representations of downstream papers, then we train a linear classifier on the downstream data to predict the field of the paper. The experimental results in Tabel 5 show that the word node embeddings are optimized as well as the paper nodes, which is the reason why our update framework works.

## 4.4 Effect of Important Word Reconstruction

In task important word reconstruction, it is true that a word may be "important" to some papers but not to others. However, even though the word decoupled from the title of a particular paper may not have a significant impact on another one, it is still a part of the overall content and meaning of that paper.

We focus on reconstructing the important words in this task which contribute to the overall semantic understanding of a particular paper. In other words, the less informative words which are not "important" to any paper can be disregarded.

| Method | Accuracy |
|---|---|
| NAR+ AWR | $65.49 \pm 0.07$ |
| NAR | $66.65 \pm 0.14$ |
| NAR+ IWR | $66.85 \pm 0.04$ |

Table 6: Effect of Important Word Reconstruction

Further, we have conducted experiments regarding to all words reconstruction(AWR) instead of important words(IWR) under Inductive learning setting, and the findings of these experiments in

Table 6 indicate that it is more effective not to reconstruct the word nodes unless we have specific preferences or criteria for word selection.

## 4.5 Ablation Studies

To verify the effects of the main components in NRUP, we further conduct several ablation studies. We choose explore under *Inductive Learning* setting.

**Effect of Update Architecture**: We explore the influence of update architecture, and table 7 shows the results that pre-train with the update architecture or not. Without the update component, we use certain self-supervised tasks for end-to-end pre-training, and keep the optimal pre-training model to generate the embeddings for downstream dataset. And in our architecture, we update the features halfway through the pre-training and retain the optimal model in the later stage. The performance on downstream dataset indicates that our framework is effective.

| Method | Accuracy |
|---|---|
| NRUP | $69.67 \pm 0.12$ |
| w/o Update Architecture | $67.56 \pm 0.03$ |
| Overall Normalization | $68.16 \pm 0.18$ |
| w/o Paper Normalization | $69.12 \pm 0.03$ |
| w/o Word Normalization | $68.54 \pm 0.07$ |
| w/o Normalization | $68.42 \pm 0.11$ |

Table 7: Effect of Update Architecture

**Effect of Normalization**: The normalization plays an crucial role in the update pre-training framework which brings the updated feature matrix back to normal distribution, eliminated the effect of distribution transfer. Table 7 shows the results that update feature without normalization and with different normalization ways. We found that the effect of normalizing the feature matrix is significantly better than not performing it. Meanwhile, the effect of normalizing the feature matrices of the papers and words separately is better than the effect of overall normalizing. In brief, normalization brings improvements.

**Effect of Adaptive Parameter**: Aggregators in NRUP are in charge of updating node features. We further explore the way and degree of aggregation by using fixed hyper-parameters instead of adaptive hyper-parameters for dataset. Figure 3 shows the results that when the value of hyper-parameters is around adaptive parameters, the effect is better.

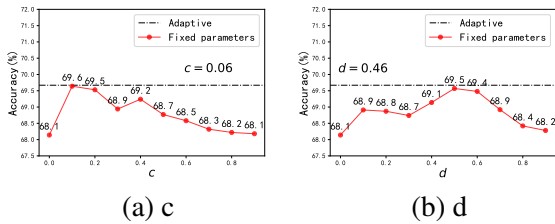

(a) c      (b) d

Figure 3: Effect of Adaptive Parameter(When exploring the hyperparameter $c$, we set $d$ as the adaptive parameter and vice versa)

## 5 Related work

**Representation Learning on Text-attributed Graphs**: Text-attributed graphs(Yang et al., 2021) are rich in semantic and structural information, previous studies on text-attributed graphs can be divided into three aspects: LMs only, GNNs only, combination of LMs and GNNs.

Early works leverage language models(Kim, 2014; Vaswani et al., 2017) to learn word representation based on sentence sequence. However, neglect of the relationship between nodes leads to underutilization of structural information. In order to leverage the interrelation of nodes more effectively, the graph's structural configuration is employed as a complementary resource alongside textual data, with the aim of augmenting the training process of language models(Yang et al., 2020; Mou et al., 2023). Besides, GIANT(Chien et al., 2021) propose to train LM with graph structure, but message passing among nodes is ignored in LM-based methods.

The development of GNNs(Kipf and Welling, 2017; Zhang and Chen, 2018) brings new ideas for studying this data format. In these works(Hu et al., 2020a), the raw text of nodes are transformed to numerical features as node attributes using graph-irrelevant methods(Mikolov et al., 2013; Devlin et al., 2019). Nevertheless, representations for text are fixed in this situation, resulting in undermining of text information.

Co-training approaches(Bi et al., 2021; Zhu et al., 2021a) with combination of LMs and GNNs have advantages of both models. However, it suffer from issues of scalability due to the size of graph and parameters of LMs. Recently, a variational EM framework(Zhao et al., 2023) propose to alternatively update the LM and GNN, but it relies on task-specific labels thus the learned representation cannot be applied to other scenario.

**Heterogeneous Graph Pre-training**: There are

studies on related pre-training methods in the field of heterogeneous graph, which leads to more general representation generated by R-GNN encoder. Jiang et al. (2021a,b) proposed two heterogeneous graph pre-training frameworks: PT-HGNN and CPT-HG, in which PT-HGNN proposed two pre-training tasks at the node level and pattern level, while CPT-HG proposed two pre-training tasks at the relation level and subgraph level, both of which achieved good results. These pre-training methods helps the model acquire the representation with generalization and effectiveness.

## 6 Conclusion

In this work, we propose to learn representations of papers and words simultaneously via co-modeling the raw text and graph based on hierarchical text-attributed graph. We design a pre-training framework and corresponding self-supervised tasks for this scenario. Sufficient experiments conducted on the benchmark dataset ogbn-arxiv demonstrate the effectiveness and generality of our method.

## Limitations

In our work, we propose to construct a hierarchical text-attributed graph to realize connections between nodes and words. However, the size of constructed heterogeneous graph is proportional to the number of initial nodes, and the memory complexity is proportional to size of graph structure. Therefore, with the increase of paper-word edges, the memory cost of NRUP may become unaffordable, which limits the scalability of our method. Meanwhile, hyper-parameters of loss function may vary in different datasets. We will manage to address these issues in future work.

## Ethics Statement

Our work strictly adheres to the ACL Ethics Policy, and the data in this paper comes from Open Graph Benchmark.

## Acknowledgements

This work is supported by National Natural Science Foundation of China (No. 6217020551) and Science and Technology Commission of Shanghai Municipality Grant (No.21QA1400600).

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

# A  Pre-training Details

This section illustrates some details during the pre-training stage.

**Update Architecture Setting**: We design the aggregators to update the paper and word features using optimized embedding. Specificly, we update the features after 2000 epochs training, and train R-GAT with the updated representations for another 2000 epochs. In Table 8, we outline the hyper-parameters of our update architecture.

| Hyper-parameters | Value |
|---|---|
| epochs | 4000 |
| optimizer | AdamW |
| hidden layer dimension | 768 |
| layers of R-GAT | 2 |
| heads of R-GAT | 4 |
| drop-out of R-GAT | 0.5 |
| activation | prelu |
| norm | layernorm |
| weight decay | 1e-4 |
| lr | 0.001 |

Table 8: Hyper-parameters of Update Architecture

**Self-supervised Tasks Setting**: In the framework, we select 2 basic self-supervised tasks: Feature Reconstruction and Link Prediction, and 2 scenario-specific tasks: Identify Important Words and Reconstruct Important Words. And we optimize the R-GAT model with the multi-loss of these tasks. Table 9 outline the hyper-parameters of self-supervised tasks.

In task node attribute reconstruction, we follow GraphMAE (Hou et al., 2022) and select a one-layer R-GNN as the decoder, because the R-GNN decoder can recover the input features of one node based on a set of paper nodes and word nodes instead of only the node itself, and it consequently helps the encoder learn high-level latent code. And the scaling cosine error can get rid of the impact of dimensionality and vector norms thus improves the training stability of representation learning.

# B  Fine-tuning Details

This section illustrates some details during the fine-tuning stage.

**Subject Prediction Setting**: In this downstream task, we leverage the embeddings output by R-GAT for prediction. We train a linear classifier on the fixed representations of downstream dataset.

| Hyper-parameters | Value |
|---|---|
| negative to positive ratio in LP | 3 |
| edge type of negative in LP | Paper-Paper |
| Decoder in FR | R-GAT |
| layers of decoder in FR | 1 |
| heads of decoder in FR | 1 |
| mask rate of FR | 0.5 |
| mask edge rate of FR | 0.5 |
| Loss function of FR | sce |
| $\gamma$ of FR | 3 |
| dimension of PH in IIW | 768 |
| layers of PH in IIW | 2 |
| mask rate of RIW | 0.2 |

Table 9: Hyper-parameters of Self-supervised Tasks

Tabel 10 outline the hyper-parameters of this task.

| Hyper-parameters | Value |
|---|---|
| epochs of linear | 2000 |
| optimizer of linear | adamW |
| lr of linear | 0.01 |
| dimension of linear | 768 |
| classification number | 40 |

Table 10: Hyper-parameters of Subject Prediction

**Important Words Identification Setting**: In this downstream task, we leverage the concatenation of paper and word embedding output by R-GAT for prediction. For baseline models, we use the concatenation of paper embedding output by GNN and initial bert embedding for prediction. We train a linear classifier on the concatenation of representations of downstream dataset. Tabel 11 outline the hyper-parameters of this task.

| Hyper-parameters | Value |
|---|---|
| epochs of linear | 1000 |
| optimizer of linear | adamW |
| lr of linear | 0.01 |
| dimension of linear | 1536 |
| classification number | 2 |

Table 11: Hyper-parameters of Important Words Identification

## C Other Experiments

In addition to the ogbn-arxiv dataset in this paper, we have conducted extra experiment on the Open Academic Graph (OAG) dataset, which contains more than 178 million paper nodes and 2.236 billion edges. Each paper is labeled with a set of research topics/fields (e.g., Physics and Medicine) and the publication date ranges from 1900 to 2019.

| Method | Paper-Field | Paper-Venue |
|---|---|---|
| GPT-GNN | $42.22 \pm 1.02$ | $46.72 \pm 0.99$ |
| GraphMAE | $43.77 \pm 0.12$ | $47.81 \pm 0.18$ |
| GIANT | $44.59 \pm 0.07$ | $48.98 \pm 0.09$ |
| **NRUP** | $\mathbf{45.01 \pm 0.13}$ | $\mathbf{49.92 \pm 0.12}$ |

Table 12: Hyper-parameters of Update Architecture

Due to our limited computing resources, we only sampled 0.5% of the nodes and corresponding edges for the hierarchical text-attributed graph construction(850000 nodes) and pre-training. For the fine-tuning step, we consider the edge prediction of Paper-Field, Paper-Venue as two downstream tasks. Table 12 shows our results.

## D The hyper-parameters for the coefficients of different losses

During the pre-training process of ogbn-arxiv, we consider node attribute reconstruction as the fundamental self-supervised task. This is because semantic information plays a crucial role in text-attributed graphs.

Also, in our experiments, we observed that setting the values of $\lambda_1, \lambda_2, \lambda_3$ (which are hyperparameters controlling the relative importance of different loss terms) to be around 0.1 to 0.2 yielded good results.

Empirical results indicate that sufficient mining of semantic information and appropriate learning of structural information can enable graph neural networks to acquire more useful transferable knowledge that is beneficial for downstream tasks.

## E Cost of Graph Construction

We run the construction process of hierarchical text-attributed graph to record the time complexity. The running time of construction process under two different settings are as follows:

- Transductive Learning: 7.5hr
- Inductive Learning: 3.5hr

The experimental results indicate that the construction step is affordable in terms of time complexity.

Furthermore, it is worth highlighting that the construction step of the hierarchical text-attributed graph only needs to be **performed once** and constructed graphs are large-scale with 10~30 million edges. Once the graph is constructed, it can be utilized for both the pre-training step and multiple downstream tasks.