# OpenReview forum: "Unleashing the Power of Language Models in Text-Attributed Graph"
_EMNLP/2023/Conference — EMNLP 2023 Findings_

### Official Review · Reviewer_TszY · 2023-08-04

**Typos Grammar Style And Presentation Improvements:** Please refer to Reasons To Reject.
**Soundness:** 3

**Excitement:**

3: Ambivalent: It has merits (e.g., it reports state-of-the-art results, the idea is nice), but there are key weaknesses (e.g., it describes incremental work), and it can significantly benefit from another round of revision. However, I won't object to accepting it if my co-reviewers champion it.

**Missing References:**

None.

**Paper Topic And Main Contributions:**

This paper focuses on text-attributed graph representation learning. The authors argue that existing works suffer from issues like underutilization of relationship between nodes or words or unaffordable memory cost. In this paper, a Node Representation Update Pre-training Architecture (NRUP) is proposed based on Co-modeling Text and Graph. In NRUP, a hierarchical text-attributed graph is constructed to incorporates both original nodes and word nodes, and four self-supervised tasks is applied for pre-training. The experimental results on ogbn-arxiv dataset demenstrate the effectiveness of the proposed method.

**Questions For The Authors:**

None.

**Reasons To Accept:**

1. The writing is clear and easy to follow.
2. The construction of hierarchical text-attributed graph is reasonable and novel.
3. The proposed pre-training framework encompasses various self-supervised tasks to ensure the generalization ability.

**Reasons To Reject:**

Minor mistakes:
1. The construction of hierarchical text-attributed graph seems to be costly.
2. Typo in Abstract: L.18 "diffferent" -> "different".
3. The numerical indices in the caption of Figure 2 are inconsistent with the display.

**Reproducibility:**

4: Could mostly reproduce the results, but there may be some variation because of sample variance or minor variations in their interpretation of the protocol or method.

**Reviewer Confidence:**

3: Pretty sure, but there's a chance I missed something. Although I have a good feel for this area in general, I did not carefully check the paper's details, e.g., the math, experimental design, or novelty.

---

> ### Author Rebuttal · Authors · 2023-08-29
>
> We thank reviewer TszY for their positive comments and valuable time. We address all concerns as follows.
>
> ### Q1: Cost of graph construction
> This is a very good question! We re-run the construction process of hierarchical text-attributed graph to record the time complexity. The running time of construction process under two different settings are as follows:
> * Transductive Learning: 7.5hr
> * Inductive Learning: 3.5hr
>
> The experimental results indicate that the construction step is affordable in terms of time complexity. Furthermore, it is worth highlighting that the construction step of the hierarchical text-attributed graph only needs to be **performed once** and constructed graphs are large-scale with 10~30 million edges. Once the graph is constructed, it can be utilized for both the pre-training step and multiple downstream tasks.
>
> ### Q2: Typo and Mismatched indices
> Thanks for pointing out the typo and the mismatched indices. We will correct them in our revision.

---

### Official Review · Reviewer_52xC · 2023-08-04

**Soundness:** 3

**Excitement:**

3: Ambivalent: It has merits (e.g., it reports state-of-the-art results, the idea is nice), but there are key weaknesses (e.g., it describes incremental work), and it can significantly benefit from another round of revision. However, I won't object to accepting it if my co-reviewers champion it.

**Paper Topic And Main Contributions:**

The main contributions of this research are as follows:
1. Constructed a hierarchical text-attributed graph that incorporates both original nodes and word nodes, which can capture the finer nuances of the text at a more granular level.
2. Introduced a multi-task graph pre-training framework that encompasses various self-supervised tasks to adapted to various scenarios.
3. Employ a relational graph neural network (R-GNN) as the foundational model for acquiring knowledge from the hierarchical text-attributed graph to mutually boost representations of nodes and words.

**Questions For The Authors:**

A.In the experimental part, the authors only selected one dataset (Ogbn-arxiv) to test the performance of the model in the downstream tasks. Can this prove the universality of the model? Why do the authors use more datasets?
B. The authors point out in the abstract that the existing work has a large memory cost problem, so how do you solve the memory cost problem? Do you need to compare the space complexity?
C. What does the column "words" mean? Why are they the same in three rows?

**Reasons To Accept:**

1.The paper introduces a hierarchical text-attributed graph construction and a multi-task graph pretraining framework, which address the underutilization of relationships between nodes and words.
2.Experimental results show the proposed method is superior to baseline methods both in terms of evaluation metrics and efficiency.



**Reasons To Reject:**

1. Only one dataset is selected for the experiment.


**Reproducibility:**

4: Could mostly reproduce the results, but there may be some variation because of sample variance or minor variations in their interpretation of the protocol or method.

**Reviewer Confidence:**

4: Quite sure. I tried to check the important points carefully. It's unlikely, though conceivable, that I missed something that should affect my ratings.

---

> ### Author Rebuttal · Authors · 2023-08-29
>
> We thank reviewer 52xC for their positive comments and valuable time. We address all concerns as follows.
> ### Q1: Other experiments
> Thanks for the suggestion! We have conducted extra experiment on the Open Academic Graph (OAG) dataset, which contains more than 178 million paper nodes and 2.236 billion edges. Each paper is labeled with a set of research topics/fields (e.g., Physics and Medicine) and the publication date ranges from 1900 to 2019.
> Due to the limited time and our limited computing resources, we only sampled 0.5% of the nodes and corresponding edges for the hierarchical text-attributed graph construction(850000 nodes) and pre-training. For the fine-tuning step, we consider the the prediction of Paper-Field, Paper-Venue as two downstream tasks. Our results are as follows:
>
> |Method|Paper-Field|Paper-Venue|
> |:------:|:------:|:------:|
> |GPT-GNN|$42.22 \pm 1.02$|$46.72 \pm 0.99$|
> |GraphMAE|$43.77 \pm 0.12$|$47.81 \pm 0.18$|
> |GIANT|$44.59 \pm 0.07$|$48.98 \pm 0.09$|
> |**NRUP**|$\bf{45.01 \pm 0.13}$|$\bf{49.92 \pm 0.12}$|
>
> The results show that NRUP achieves better performance compared to the competitive  baselines, demonstrating its effectiveness on other dataset. We will update the final result in our revision in the future.
> ### Q2: Memory cost problem
> Thanks for the suggestion! Existing work suffers from the memory cost problem due to training language model(LM) or co-training LM and GNN in the pre-training step. However, in our NRUP, we use a pre-trained LM to generate the initial embedding for the word nodes and original nodes, and we exclusively rely on graph neural networks (GNN) during pre-training, which significantly reduces the number of parameters compared to LM-based methods.
> The parameters of different methods are as follows:
>
> |Method|Params|
> |:------:|:------:|
> |GPT-GNN|456192|
> |GraphMAE|300288|
> |GIANT|>7000000|
> |**NRUP**|748032|
>
> Note that the parameters of our method NRUP is slightly larger than GNN-based methods due to the additional word nodes, but significantly less than LM-based methods GIANT. It shows that our method is affordable in terms of space complexity.
> ### Q3: The meaning of "words"
> In table 2, the column "words" indicates the number of word nodes in hierarchical text-attributed graph. To ensure the generality and effectiveness of our method, we use the **same tokenizer** to decouple words from the original text, which means words decoupled from the corpus are **shared**. Therefore, there is an equal number of word nodes in all three graphs.

---

### Official Review · Reviewer_y7tX · 2023-08-05

**Soundness:** 3

**Excitement:**

3: Ambivalent: It has merits (e.g., it reports state-of-the-art results, the idea is nice), but there are key weaknesses (e.g., it describes incremental work), and it can significantly benefit from another round of revision. However, I won't object to accepting it if my co-reviewers champion it.

**Paper Topic And Main Contributions:**

This papers proposes a pre-training framework for learning representations of nodes and words simultaneously in a text-attributed graph. The proposed framework, called NRUP, first constructs a hierarchical test-attributed graph where a set of nodes is composed of text elements (e.g., papers in citation networks) and words and they are connected to each other via citations or co-occurrences. The node representations are learned via four different self-supervised tasks. During pre-training, they are further optimized by aggregation and normalization. After pre-training, they are fine-tuned to perform downstream tasks. To demonstrate its effectiveness, the ogbn-arxiv dataset is used and two different downstream tasks are used for evaluation. The experimental results show the proposed method can achieve competitive performances compared to the baseline methods.

**Questions For The Authors:**

* Are there experiment results on other dataset or other downstream tasks?
* In Important Word Reconstruction, a word may be important to some papers but not to others. Wasn't it an issue when pre-training?
* The hyper-parameters for the coefficients of different losses need to be clarified.
* No edge weights are used. Any reasons?

**Reasons To Accept:**

* Overall, the paper is easy to read (although there are some confusing terms like "original/initial nodes").
* The proposed framework can be generally applied to various LM and GNN models.

**Reasons To Reject:**

* It is unclear whether the proposed framework is actually effective to capture the co-dynamics of graph structure and text attributes. The pre-training objectives seem to heavily rely on the textual attributes, and the second-order relationships of nodes are not considered at all, which has been known as very important to be able to well understand graph structure. Also, the two downstream tasks used for evaluation are related to only node attributes. There needs another tasks that evaluate on graph structure as well.
* Only one dataset is used in the experiment, which makes the proposed idea sound less convincing.
* The experiments do not seem fair. None of the baselines use LM. At least, one method that uses LM and GNN should have been compared together. Also, The downstream tasks are sort of replaying some of the objectives used in NRUP.

**Reproducibility:**

3: Could reproduce the results with some difficulty. The settings of parameters are underspecified or subjectively determined; the training/evaluation data are not widely available.

**Reviewer Confidence:**

4: Quite sure. I tried to check the important points carefully. It's unlikely, though conceivable, that I missed something that should affect my ratings.

**Typos Grammar Style And Presentation Improvements:**

In Figure 2, the figure numbers do not match their corresponding descriptions. (e.g., "2. Self-supervised Tasks" should be "3. Self-supervised Tasks")

---

> ### Author Rebuttal · Authors · 2023-08-29
>
> We thank reviewer y7tX for their positive comments and valuable time. We address all concerns as follows.
>
> ### Q1: Other experiments
> Thanks for the suggestion! We have conducted extra experiment on the Open Academic Graph (OAG) dataset, which contains more than 178 million paper nodes and 2.236 billion edges. Each paper is labeled with a set of research topics/fields (e.g., Physics and Medicine) and the publication date ranges from 1900 to 2019.
> Due to the limited time and our limited computing resources, we only sampled 0.5% of the nodes and corresponding edges for the hierarchical text-attributed graph construction(850000 nodes) and pre-training. For the fine-tuning step, we consider the edge prediction of Paper-Field, Paper-Venue as two downstream tasks. Our results are as follows:
>
> |Method|Paper-Field|Paper-Venue|
> |:------:|:------:|:------:|
> |GPT-GNN|$42.22 \pm 1.02$|$46.72 \pm 0.99$|
> |GraphMAE|$43.77 \pm 0.12$|$47.81 \pm 0.18$|
> |GIANT|$44.59 \pm 0.07$|$49.28 \pm 0.09$|
> |**NRUP**|$\bf{45.01 \pm 0.13}$|$\bf{49.92 \pm 0.12}$|
>
> The results show that our NRUP achieves better performance compared to the competitive  baselines, demonstrating its effectiveness on a different dataset. Also, these downstream tasks are evaluated on **link-level(graph structure)**. We will update the final result in our revision in the future.
>
> ### Q2: Baseline of LM and GNN
> First, we respectfully disagree with the comment that "None of the baselines use LM". In fact, baseline GIANT uses LM for pre-training. And our approach NRUP achieves competitive or even better performance compared to that.
> Second, the reason why we didn’t use the method that uses the combination of LM and GNN is that such an approach is effective when graphs are small. However, it becomes problematic once the graphs become very large.
> This is because the size of the computational graph is proportional to the size of the graph structure between nodes as well as the language model capacity, and the memory complexity is proportional to the size of the observed graph and the number of parameters in the LM. So this kind of method is not suitable under limited computing resources.
>
> ### Q3: About “The downstream tasks are sort of replaying some of the objectives used in NRUP.”
> It is worth emphasizing that we avoid performing our self-supervised tasks on the test part of downstream tasks in the pre-training step, ensuring that testdata objectives have not appeared during the pre-training step. In transductive learning, NRUP does not perform the task “Important Word Identification” on the test portion of downstream task; and in inductive learning, our approach involves pre-training and fine-tuning on two distinct graphs.
> It is common that downstream tasks share the same objectives with pre-training step in real world applications. For example, downstream link prediction task in biology$^{[1]}$ or chemistry$^{[2]}$ often involves link reconstruction during pre-training$^{[3]}$. Massive unlabeled data in pre-training provide inherent structure information in graphs.
>
> > [1] Stanfield, Z., Coşkun, M. & Koyutürk, M. Drug Response Prediction as a Link Prediction Problem. Sci Rep 7, 40321 (2017). https://doi.org/10.1038/srep40321
> [2] Lu, Y., Guo, Y. & Korhonen, A. Link prediction in drug-target interactions network using similarity indices. BMC Bioinformatics 18, 39 (2017). https://doi.org/10.1186/s12859-017-1460-z
> [3] Hu, Weihua, et al. "Strategies for pre-training graph neural networks." arXiv preprint arXiv:1905.12265 (2019).
>
>
> ### Q4: Issue of Important Word Reconstruction
> This is a very good question! In this task, it is true that a word may be "important" to some papers but not to others. However, even though the word decoupled from the title of a particular paper may not have a significant impact on another one, it is still a part of the overall content and meaning of that paper.
> We focus on reconstructing the important words in this task which contribute to the overall semantic understanding of a particular paper. In other words, the less informative words which are not "important" to any paper can be disregarded.
> Further, we have conducted experiments regarding to all words reconstruction instead of important words under the same setting in Section 4.2, and the findings of these experiments indicate that it is more effective to not reconstruct the word nodes unless we have specific preferences or criteria for word selection.
> |Method|Accuracy(inductive learning)|
> |:------:|:------:|
> |NAR+ All Word Reconstruction|$65.49 \pm 0.07$|
> |NAR|$66.65 \pm 0.14$|
> |NAR+ Important Word Reconstruction|$66.85 \pm 0.04$|
>
> ### Q5: The hyper-parameters for the coefficients of different losses.
> Thanks for the question! In our method, we consider node attribute reconstruction as the fundamental self-supervised task. This is because semantic information plays a crucial role in text-attributed graphs.
> Also, in our experiments, we observed that setting the values of $\lambda_{1},\lambda_{2},\lambda_{3}$ (which are hyperparameters controlling the relative importance of different loss terms) to be around 0.1 to 0.2 yielded good results.
> We will include an explanation of hyper-parameters coefficients in the revision.
>
> ### Q6: Edge weights
> This is a very good question! We agree that edge weights often provide important information about the strength or significance of relationships between nodes. However, in the case of the ogbn-arxiv dataset, there is no available information on edge weights. More specifically, the dataset does not provide any specific measurements or values indicating the strength or intensity of the "cite" relationship between papers.
> Since this information is not provided, it becomes difficult for us to accurately assign meaningful weights to the edges.
> As a result, we refrained from using edge weights in our method. We will further explore datasets with edge weights information and verify the effectiveness of our method on them.

---

### Meta-Review · Area_Chair_H4B8 · 2023-09-15

**Recommendation:** 4

**Metareview:**

The paper proposes a pretraining framework for learning representations of nodes in text-attributed graphs, applicable to GNNs and other models of structured data. The authors demonstrate that their proposed approach is competitive with or marginally better than previous approaches, using node classification in scientific graphs as the test case.

The primary flaw identified by reviewers was the decision by the authors to evaluate on only one of the standard benchmarks for the task. This weakened the empirical arguments in the paper. Since the initial submission, the authors have in their rebuttal provided another set of results, significantly strengthening the paper.

As the framework is quite general and appears to yield performance improvements (albeit only to a small degree), the method presented here may find wider applicability.

---

### Decision · Program_Chairs · 2023-10-07

**Decision:**

Accept-Findings

**Comment:**

The paper proposes a pretraining framework for learning representations of nodes in text-attributed graphs, applicable to GNNs and other models of structured data. The authors demonstrate that their proposed approach is competitive with or marginally better than previous approaches, using node classification in scientific graphs as the test case.

The primary flaw identified by reviewers was the decision by the authors to evaluate on only one of the standard benchmarks for the task. This weakened the empirical arguments in the paper. Since the initial submission, the authors have in their rebuttal provided another set of results, significantly strengthening the paper.

As the framework is quite general and appears to yield performance improvements (albeit only to a small degree), the method presented here may find wider applicability.